# Heterogeneous Winter Wheat Populations Differ in Yield Stability Depending on their Genetic Background and Management System

**Odette D. Weedon** *[ID] and **Maria R. Finckh**[ID]

Department of Ecological Plant Protection, University of Kassel, Nordbahnhofstr. 1 a, 37213 Witzenhausen, Germany; mfinckh@uni-kassel.de
* Correspondence: odetteweedon@uni-kassel.de

**Abstract:** Twelve winter wheat composite cross populations (CCPs), based on three genetic backgrounds and maintained at the University of Kassel, Germany, under both organic and conventional management, were assessed for yield performance and stability in comparison to two commercial varieties over eight and 10 experimental years. A number of stability parameters were chosen in order to identify populations with either adaptation to specific environments or broad adaptation across environments. The genetic effects of the CCP parental varieties were clearly present when comparing CCP yield performance in both management systems. Compared to the variety 'Capo', CCPs yielded similarly under organic, but poorer under conventional conditions. Under both management systems, CCPs with the broadest or with a more modern (high yielding) genetic base achieved the greatest yield stability, exceeding that of 'Capo', and demonstrating the buffering capacity of genetic diversity. CCPs with a genetic background of high yielding parents reacted most strongly to the different environments and apparently diverged under conventional management over time. Possibilities to improve CCPs through the addition of new genetic material while maintaining the benefits of diversity to achieve higher and more stable yields, particularly in light of increasingly unpredictable climatic conditions are discussed.

**Keywords:** heterogeneous populations; winter wheat; yield stability; organic and conventional management; genetic diversity

## 1. Introduction

Increasingly uncertain climatic conditions threaten the security and stability of agricultural systems [1] and a better understanding of genotype × environment interactions (GEIs) is needed to shift the agricultural focus from manipulating environments to grow crops to creating crops that fit into the environment [2,3].

Environmental interactions, besides climatic and soil effects, include the effects of agricultural management systems [3]. GEIs play a pivotal role in assessing the yield stability of crops and the challenge of changing climatic conditions necessitates that new crop cultivars should have broad adaptability, stable agronomic performance over a range of environments and management systems and generally high yields [4]. However, there is an on-going debate as to whether the currently used diversity within cereals is sufficient to allow for a great enough response diversity to climatic variability [5–8]. As an alternative, intraspecific diversification has been advocated as a means to enhance adaptability to unpredictable environments [1,2,9–11]. Provisions have been made to allow for the use of more intraspecific diversity in Europe [12,13], a development of greater interest for the organic sector.

Yield stability is an important selection criterion in varieties and plant material suited for organic and low-input agricultural systems, particularly due to the challenging environmental conditions found in these systems [14–16]. Genotypes with better yield stability may have an advantage under such conditions [17,18], particularly as GEIs are considered "a common limiting factor when stable genotypes are required" [19].

A number of yield stability parameters are available in order to measure stability in crops. Yield stability is divided into a couple of concepts, the main two being the concept of static stability (Type I, homoeostasis, biological) [20] and dynamic stability (Type II, agronomic) [21]. The static stability concept refers to genotypes, which generally tend to yield similarly across all environments and are thus relatively better yielding in unfavorable environments [22]. According to Simmonds [23], genotypes that show high stability in the static sense may be better suited to organic and low-input agricultural systems. Static stability parameters enable the selection of genotypes better suited to marginal environments characterized by high environmental stresses, which makes this stability concept particularly relevant for small-scale, subsistence farmers [24]. The static stability concept makes use of specific GEIs, in order to find genotypes that are better suited to a given environment, thereby selecting for "more narrowly adapted varieties" [25]. The dynamic concept of stability refers to a genotype whose mean response in an environment is parallel to the mean response of all genotypes in an environment. These genotypes tend to have low GEIs and are adaptable to a broader range of environments [17,25], which may also be an important characteristic valuable to organic agricultural systems depending on the range of environments. Plant breeding for conventional agriculture, for example, has focused on dynamic yield stability in order to produce varieties better adapted to diverse geographical environments and suited to less challenging environments that can be manipulated or managed through higher inputs [26], so that genotype yield improves through improved environmental conditions [27].

A number of researchers have reported on the advantages of increased diversity, both inter- and intraspecific, for plant protection, yield stability, and soil conservation [1,9–11]. Furthermore, positive correlations between diversity and stability, as well as between diversity and productivity have been found [28–32]. Genetic diversity is needed to be able to react and adapt to highly unpredictable climatic conditions and changes and variations in soil, water and nutrient status [2]. The High genetic Diversity approach is relevant to organic and low-input agricultural systems due to the comparatively large environmental variability found in these systems, where increased genetic diversity is better able to cope with higher biotic and abiotic stresses [14,16,33].

These challenges, coupled with the additional pressure of plant genetic diversity loss, has driven novel breeding approaches such as composite cross populations (CCPs) and other genotype mixtures, thereby increasing intra-field diversity. Literature on the effects of increased genetic diversity through the use of cultivar mixtures and CCPs and its stabilizing effect on yield has, however, been contradictory. Hockett et al. [34] reported on inconsistent yield stability for a number of barley populations analyzed across 16 environments. Clay and Allard [35], working with barley varietal mixtures, reported that there was either no effect of increased diversity on yield stability or that although varietal mixtures resulted in a small yield advantage over the mean of their components, yield stability was found to be lower. However, a number of researchers report on greater yield stability in more genetically diverse crop material in both barley [19,29,36–38] and wheat [39,40]. Positive effects of genetically diverse cropping systems on yield stability in general have also been reported [41,42]. However, little is known about the yield stability of heterogeneous wheat populations and how to adequately describe and compare the yield stability across growing systems.

In this paper, we present the results of a yield stability analysis on a number of winter wheat CCPs and pure line varieties that were grown under both organic and conventional management for up to 10 experimental years. In 2002, three winter wheat CCPs were created through the crossing of 20 carefully selected parental varieties by the John Innes Centre and in collaboration with the Elm Farm Research Centre, UK [40,43,44]. Since 2005, these three heterogeneous winter wheat populations have

been multiplied annually at the University of Kassel, Neu-Eichenberg, under differing management systems (organic and conventional) and subject to natural selection. At least one reference pure line variety was grown alongside the populations every year. Kang [45] recommended more than eight environments in order to calculate yield stability reliably. Under organic management, data from 10 and under conventional management, eight experimental years are available, so that a number of yield stability indices for the CCPs and reference varieties could be reliably calculated and compared. The specific aims of the research were to test: (i) the suitability of the genetically diverse CCPs in terms of yield and yield stability in the differing management systems and (ii) to compare the yield and yield stability of the CCPs to the modern pure line varieties used as references under differing management conditions.

## 2. Materials and Methods

### 2.1. Experimental Material and Field Site

In 2002, three composite cross populations (CCPs) were created in the UK through the complete half-diallel crossing of 20 parental varieties. The parental varieties (varietal release dates between 1934 and 2000) were chosen based on their agronomic performance under low-input conditions in Europe, as well as to ensure a wide genetic basis for the CCPs [40,43,44,46]. The first CCP was created through the crossing of 12 high baking quality parental varieties, thereafter referred to as CCQ (Composite Cross Quality). The second CCY was based on the crosses of nine high yielding parental varieties (Composite Cross Yield) with the parental variety Bezostaya, regarded as both high yielding and as having good baking quality, having been crossed into both the CCQ and CCY populations. The third CCP was created through the crossing of 8Y × 11Q parental varieties and Bezostaya with all 19 others and is therefore referred to as CCYQ (Composite Cross Yield × Quality). Four male sterile lines were included in the initial crosses in order to encourage outcrossing within the CCPs and the $F_2$ seeds of these initial crosses were bulked and multiplied at four sites across the UK until the $F_4$ (for detailed descriptions see [40,43,44]). Since 2005 ($F_5$), the three CCPs have been grown at the experimental research station of the University of Kassel in Neu-Eichenberg, Hessen, under both organic (O) and conventional (C) management. In 2006/2007, each CCP was split in equal parts in order to create two parallel populations per CCP under each management system (six CCPs per system), bringing the total number of CCPs to 12 (parallel populations indicated by I and II).

The 12 CCPs have been multiplied annually in Neu-Eichenberg since 2006/2007 at the experimental research fields of the Department of Ecological Plant Protection (51°22′24.7″ N and 9°54′12.5″ E, 247 m above sea level). Mean annual precipitation taken always from September to August reflecting the winter wheat season from 2005–2018 was 664 mm and mean annual temperature of 9.6 °C (Table A1). Plot size was at least 150 m² from which multiple replicate yield samples were taken annually. The organically managed CCPs were grown in all years in a field (Teilanger) classified as having fine loamy loess soil (Haplic Luvisol) with 76 soil points according to the German soil grading system (0–100) [47]. From 2005/2006 to 2012/2013, the conventional CCPs were grown on a field (Saurasen) located at about 500 m distance from Teilanger in a sandy loam loess soil (Stagnic Luvisol) with 52 soil points [47]. Since 2013/2014, the conventional CCPs were relocated to a conventionally managed field on Teilanger, thereby ensuring similar soil quality for both management systems.

The organic CCPs have been managed without the addition of fertilizers, fungicides, pesticides, or herbicides and follow a two-year grass-clover mixture. Row spacing was generally 30 cm to allow for weed control through harrowing at tillering and/or hoeing at later stages if needed (see [48] for further details). The conventional populations on Saurasen were rotated annually with grass-clover and the pre-crop for the conventionally managed CCPs on Teilanger was either maize or a green manure. No fungicides or insecticides were applied, but herbicide was generally used once a year in early spring. A split application of mineral nitrogen fertilizer at a rate of 125 kg/ha was applied during the growing season (50 kg and 75 kg/ha at each application, respectively). At stem elongation,

growth regulators were applied in 2008 and 2009, but not in 2010 or thereafter. A number of currently relevant pure line varieties were grown in smaller plots neighboring the CCPs for comparison in both the organic and conventionally managed systems. Under organic management, the reference varieties 'Achat' and 'Capo' were grown in 10 experimental years (2007/2008–2017/2018) with the CCPs. However, under conventional management only the variety 'Capo' was used most often as a reference variety since 2008/2009 (eight experimental years). Both 'Achat' and 'Capo' are commonly grown varieties known for baking quality (E class varieties according to the German wheat classification system) (see [49] for details on varieties).

## 2.2. Stability Parameters and Statistical Analysis

Stability indices based on grain yield values for the CCPs and reference varieties were calculated separately for each management system. The CCPs are referred to as CCP entries when referred to as a group or by their individual names (e.g., CYQI—conventionally managed CCYQ parallel population I) and the reference varieties by their names 'Achat' and 'Capo'. The experimental years are treated as environments (Env.) and the CCP entries and reference varieties are referred to as genotypes (Gen.) when a genotype × environmental interaction (GEI) is discussed. The yield stability indices of the organically managed CCPs and 'Achat' and 'Capo' were calculated based on a dataset of 10 experimental years (2007/2008–2011/2012 and 2013/2014–2017/2018), whereas the dataset of the conventional CCPs and 'Capo' was based on eight experimental seasons (2008/2009–2009/2010 and 2012/2013–2017/2018). Grain yield in each management system was analyzed separately using a Linear Mixed Effects Model using the statistical software R [50] and the R packages *lme4* [51], *nlme* [52], and *emmeans* [53]. The main effects included the experimental entries (Gen.) and years (Env.) and the interaction between them as fixed factors and replicate yield samplings per year as the random effect, whilst in the conventional system, location was additionally included as a random effect. Year (experimental season) was used as a fixed factor for each separate growing system as year and generation are inseparable for the CCPs. Thus year effects should be seen as year × generation effects [44]. For all analyses, fixed and random factor interactions were included or eliminated depending on the model with the lowest Akaike Information (AIC) and Bayesian Information (BIC) Criterion fitted by the maximum log-likelihood. Factors of the final model were then tested for violation of variance homogeneity using the Levene Test and fitted by maximizing the restricted log-likelihood. For models indicating heteroscedastic distribution of residuals, the constant variance function 'varIdent' in the *emmeans* package [53] was included in the model for each treatment [54].

Environmental variance ($EV_i$) as a stability index falls under the Type I stability concept of static stability. Genotypes that achieve low $EV_i$ values are considered stable as their grain yields show very little variation across environments and hence no strong GEI [20,55]. However, genotypes that indicate good static stability by achieving constant yields across environments may yield better under challenging environments indicating suitability to specific environments. The stability index environmental variance ($EV_i$) was calculated by:

$$EV_i = \sum \frac{(X_{ij} - X_i)^2}{(e-1)},\tag{1}$$

with $X_{ij}$ representing the yield response of the entry $i$ in the environment $j$, $\overline{X}_i$ as the entry mean yield across all environments, and $e$ representing the number of environments.

The yield reliability index ($I_i$) proposed by Kataoka [56] combines both genotype mean yield and $EV_i$ to combine the advantages of genotypes with better grain yield and low yield variation across environments. The index calculates the lowest yield that a genotype will achieve based on a probability (p), which is fixed and dependent on a farmer's aversion to risk or a particular environment [17]. Döring et al. [40] used the fixed $p = 0.75$ in order to calculate $I_i$ for a number of CCPs and their mixtures. This p value allows for the lowest yield that could be expected in 75% of cases and has been

recommended for modern agricultural systems with favorable climatic conditions [17,40,57]. $I_i$ was calculated by:

$$I_i = X_i - Z_{(p)} \sqrt{EV_i}, \tag{2}$$

where $Z_{(P)}$ equals the fixed factor 0.675 based on $p = 0.75$, $X_i$ represents the entry mean across all environments and $EV_i$ is the previously calculated environmental variance.

Wricke's ecovalence ($W^2$) [58] is a measure of dynamic stability (Type II stability), where a stable genotype shows a low GEI by achieving a yield response similar or parallel to the mean yield response of all genotypes [17]. This index measures the GEI effect of genotypes across environments with an index value closest to 0, indicating the highest stability with low GEI. Wricke's ecovalence $W^2$ was calculated by:

$$W^2 = \sum_{ij} \left(X_{ji} - \overline{X}_i - \overline{X}_j + \overline{X}\right)^2 \tag{3}$$

where $X_{ij}$ represents the genotype (*i*) yield in environment *j*, $\overline{X}_i$ is the genotype mean yield across environments, $\overline{X}_j$ is the mean yield in environment *j*, and $\overline{X}$ is the grand mean.

A genotype superiority index ($P_i$) was proposed by Lin and Binns [59], which assesses a genotype's yield performance in comparison to the maximum yield response in multiple environments. Lower values for $P_i$ indicate yield performance superiority, an indication that a genotype's yield response is close to the maximum yield achieved in an environment and as such is indicative of general adaptability. $P_i$ was calculated by:

$$P_i = \sum \frac{(X_{ij} - M_j)^2}{2e} \tag{4}$$

where $X_{ij}$ is the genotype (*i*) yield response in environment *j*, $M$ is the maximum/best response in environment *j*, and *e* the number of environments. $P_i$ can be extrapolated further into:

$$P_i = [e(\overline{X}_i - \overline{M})^2 + \sum_{j=1}^{e} (X_{ij} - \overline{X}_i - \overline{M}_j + \overline{M})^2]/(2e), \tag{5}$$

where $\overline{X}_i$ is the mean yield of genotype *i* over all environments and $\overline{M}$ is the mean of the maximum responses over all environments. The first part of numerator before the summation calculates the sum of squares (SS) for the genetic effects of $P_i$ (genetic deviation), whilst the second part of the expanded Equation (the summation) gives the SS of the GEI in the comparison of differing genotypes (GEI deviation) [59]. According to Lin and Binns [59], the GEI deviation should not be used as a selection criterion for stability per se, but should rather be used as an additional tool to assess specific environmental adaptation.

Additionally, linear regression of the mean yield performance of each entry on the mean environmental yield performance (mean of all experimental entries per year) was performed in order to calculate the regression coefficient ($b_i$) [60], as well as the deviation from the regression line (MSE) [27].

An AMMI (Additive Main Effect and Multiplicative Interaction) analysis was performed on the organically managed CCPs and reference varieties on a dataset of 10 experimental years (Env.) and on a dataset of eight experimental years for the conventionally managed CCPs and reference variety. The AMMI analysis combines an ANOVA (analysis of variance analysis) for the genotype and environment main effects with principal component analysis [61–63] and was carried out with the statistical software R using the AMMI function in the R package *agricolae* [64]. AMMI analysis allows for the quantification of the GEI and the visualization of patterns and relationships between genotype entries and environments through the AMMI Biplot [63]. The ANOVA allows for the compartmentalization of the yield variance into three components, namely the genotype, environment and GEI deviations from the grand or model mean. Thereafter multiplication effect analysis separates the GEI deviations into different interaction principal component axes (IPCA) [65]. For the ANOVA calculation, the main effects of experimental entries (Gen.) and experimental years (Env.) were used

as fixed factors, as well as the interaction between them for grain yield values, whilst the replicate samplings were included as the random effect.

Additionally, using the AMMI function in the R package *agricolae* [64], the AMMI stability value (ASV), proposed by Purchase et al. [66], provides a quantitative stability value based on the AMMI model and is calculated by:

$$ASV = \sqrt{\left[\frac{IPCA1_{sum\ of\ square}}{IPCA2_{sum\ of\ square}}\left(IPCA1_{score}\right)\right]^2 + \left(IPCA1_{score}\right)^2} \tag{6}$$

As described by Farshadfar et al. [65], ASV "is the distance from zero in a two dimensional scattergram of IPCA1 (interaction principal component analysis axis 1) scores against IPCA2 scores". Genotypes with lower ASV values are considered more stable due to their closer proximity to the centre of the axes and their lower GEIs (stability across environments). Genotypes that achieve low ASV values may be considered more stable; however, their yields may be low. An additional yield stability index ($YS_i$) could be calculated using the ranking of ASV values combined with yield ranking (*RS*), as proposed by Kang [67]. This yield stability index allows for the selection of genotypes characterized by both high stability and yield by ranking genotypes in terms of their ASV values and mean grain yield from best (1) to worst and summing the two ranks to provide the yield stability index ($YS_i$) [65]. $YS_i$ was calculated with the following formula:

$$YS_i = RASV + RY, \tag{7}$$

where *RASV* represents the ASV ranking and *RY*, the yield ranking. Genotypes that achieve low $YS_i$ values indicate varieties that shown stability across environments whilst achieving high yields.

Spearman rank correlation coefficients were calculated between grain yield and all stability indices combining data from both management systems (organic and conventional) using The R Stats Package *stats* (version 3.5.3) [50].

## 3. Results

### 3.1. Weather Data

The mean annual temperature (Sept.–Aug.) from 2005/2006 to 2017/2018 was 0.6 °C higher compared to the long-term annual average temperature of 9 °C recorded from 1971 to 2000. Five seasons were >10 °C (2006/2007, 2013/2014, 2014/2015, 2015/2016, and 2017/2018), whilst the coolest experimental season was 2012/2013 (8.4 °C) (Table A1). Overall, mean precipitation per month was similar to the long-term mean. The increased temperatures, however, led to higher overall evaporation. Additionally, in contrast to earlier times, when rainfall distribution was typically well balanced within months, the recent past has been characterized by relatively long dry spells for the region lasting between 6 and 12 weeks and unusually heavy torrential rains resulting sometimes in run-off and partial unavailability of the water for the plants. During the seasons that were included in the stability analyses the wettest season recorded was in 2009/2010 (791 mm total ppt.) (Table A1). Severe water stress occurred during 2014/2015 (589 mm) and 2017/2018 (471 mm) In both, 2014/2015 and 2017/2018, drought conditions were extended from spring and into the summer months (May to July), resulting in very dry seasons, particularly in 2017/2018, which also included an exceptionally dry February and extremely hot temperatures from April onwards (Table A1). Another extreme event occurred in February of 2011/2012 when after a frost-free winter a sudden and severe two-week black frost resulted in frozen soil up to 50 cm in depth. This was followed by above average temperatures during which the soil slowly thawed and little precipitation in February and March. These conditions led to severe drought stress and winterkill (see [43,44] for more details).

### 3.2. Organic CCPs and Reference Varieties

There were no significant yield differences between the organically managed CCPs and the two reference varieties (Table 1a). Across all seasons, the OYII CCP achieved the highest yield (4.93 t/ha), followed by the two reference varieties 'Achat' and 'Capo'. Both OQ CCPs were the lowest yielding entries over the 10 experimental years. In terms of environmental variance ($EV_i$), indicative of the static stability concept, the OYI and OQI CCPs achieved the lowest values (1.44 and 1.46, respectively), indicating best stability, followed by the reference variety 'Capo' (Table 1a), while 'Achat' had the highest $EV_i$ (1.98), indicating high yield variance over all environments. The highest yield reliability $I_i$, which combines both mean grain yield and $EV_i$, was achieved by the entries with the best yields, the OYII CCP, and the reference variety 'Achat' ($I_i$ = 4.37 and 4.35, respectively), confirming the strong influence of mean grain yield on this stability parameter. The OYQII and OYQI CCPs achieved the lowest ecovalence ($W^2$) values, indicating better dynamic stability in comparison to the reference varieties and the other CCP entries. These results are supported by similar rankings for MSE (mean square error) values and for ASV (AMMI Stability Value) (Table 1a).

**Table 1.** Estimated marginal means of grain yield (*GY*), environmental variance ($EV_i$), Wricke's ecovalence ($W^2$), yield reliability index ($I_i$), mean square error (MSE), AMMI (Additive Main Effect and Multiplicative Interaction) stability value (ASV), yield stability index (YS$_i$) and superiority index ($P_i$) with its genetic and genotype × environment interactions (GEI) deviations and percentual contribution of the deviations to $P_i$ for (a) the organically managed and (b) the conventionally managed composite cross populations (CCPs) and reference varieties over 10 and eight experimental years, respectively. Differing letters indicate significant differences at $p < 0.05$ (Linear mixed effects model with pairwise comparison and Holm correction). The genotypes ranked in the top three for each stability index (not including $b_i$) are marked in bold and TOP 3 is the frequency that genotypes have appeared in the top three rankings over all included stability indices.

| Entry | GY | $EV_i$ | $W^2$ | $I_i$ | $b_i$ | MSE | ASV | YS$_i$ | $P_i$ (x1000) | $P_i$ genetic | $P_i$ GEI | $P_i$ gen % | $P_i$ GEI % | TOP 3 |
|---|---|---|---|---|---|---|---|---|---|---|---|---|---|---|
| | | | | | **(a) Organic CCPs and reference varieties** | | | | | | | | | |
| Achat | **4.85** | ab | 1.98 | 1.85 | **4.35** | 1.07 | 0.22 | 1.71 | **9** | **130.3** | 69.6 | 60.7 | 53 | 47 | 4 |
| Capo | **4.87** | ab | **1.55** | 1.38 | **4.32** | 0.95 | 0.17 | 1.41 | 10 | 224.4 | 130.2 | 94.2 | 58 | 42 | 3 |
| OQI | 4.44 | b | **1.46** | 0.96 | 3.97 | 0.94 | 0.11 | **0.55** | 11 | 448.3 | 393.3 | 55.0 | 88 | 12 | 2 |
| OQII | 4.51 | b | 1.72 | **0.84** | 4.03 | 1.02 | **0.10** | 0.72 | 11 | 380.9 | 284.1 | 96.8 | 75 | 25 | 2 |
| OYI | 4.74 | ab | **1.44** | 1.01 | 4.19 | 0.93 | 0.12 | 1.16 | 11 | 336.8 | 222.7 | 114.1 | 66 | 34 | 1 |
| OYII | **4.93** | a | 1.73 | 1.65 | **4.37** | 1.00 | 0.21 | 1.62 | **9** | **174.1** | 88.1 | 86.0 | 51 | 49 | 4 |
| OYQI | 4.83 | ab | 1.72 | **0.76** | 4.23 | 1.03 | **0.09** | **0.24** | **6** | 244.7 | 153.8 | 90.9 | 63 | 37 | 4 |
| OYQII | 4.83 | ab | 1.81 | **0.48** | **4.32** | 1.07 | **0.05** | **0.30** | **5** | **123.4** | 97.1 | 26.4 | 79 | 21 | 6 |
| | | | | | **(b) Conventional CCPs and reference variety** | | | | | | | | | |
| Capo | **5.67** | a | 0.65 | 1.16 | **5.22** | 0.94 | 0.19 | 0.97 | 7 | **53.6** | 8.7 | 44.9 | 16 | 84 | 4 |
| CQI | 4.87 | e | **0.41** | **0.32** | 4.54 | 0.84 | **0.04** | **0.31** | 9 | 443.9 | 421.1 | 22.8 | 95 | 5 | 4 |
| CQII | 5.01 | de | 0.63 | **0.65** | 4.56 | 0.99 | 0.11 | **0.53** | 9 | 365.0 | 314.2 | 50.8 | 86 | 14 | 2 |
| CYI | 5.07 | cd | 1.24 | 1.40 | 4.47 | 1.45 | **0.10** | 1.17 | 12 | 401.0 | 227.0 | 174.0 | 57 | 43 | 1 |
| CYII | **5.35** | b | **0.56** | **0.07** | **4.95** | 1.00 | **0.01** | **0.17** | **3** | **126.8** | 95.8 | 31.0 | 76 | 24 | 8 |
| CYQI | **5.37** | b | **0.43** | 0.89 | **4.97** | 0.78 | 0.12 | 0.72 | 7 | **164.0** | 115.5 | 48.5 | 70 | 30 | 5 |
| CYQII | 5.23 | bc | 0.68 | 1.03 | 4.80 | 0.99 | 0.17 | 0.87 | 9 | 235.9 | 141.2 | 94.6 | 60 | 40 | 0 |

The genotypes ranked in the top three for each stability index (not including $b_i$) are marked in bold and TOP 3 is the frequency that genotypes have appeared in the top three rankings over all included stability indices.

The best superiority index ($P_i$) value was found in the OYQII CCP (123.4), followed by 'Achat' (130.3) and OYII (174.1) (Table 1a). Lower $P_i$ values indicate smaller distances from maximum yields and therefore "greater yield superiority" [40]. However, the contribution of Genotype-Environment Interactions (GEI) to $P_i$ for 'Achat' and OYII (47% and 49%, respectively) was considerably greater than that of OYQII (21%). The greater contribution of GEI deviation to $P_i$ for 'Achat' and OYII indicates that these entries show a likelihood towards greater GEI interaction and adaptation to specific environments. This is supported by the greater ecovalence ($W^2$) and ASV values for these two entries

(Table 1a). The higher $P_i$ values of OQI, OQII, and OYI indicate lower yields in comparison to the other entries; however, the genetic contribution to their $P_i$ was considerably greater indicating more general adaptation, particularly as the $W^2$ values of these entries were also low. In terms of stability based on the genotype regression coefficient ($b_i$) [60], the OYII CCP was equally adapted to all environments with a slope ($b_i$) of 1. The OQI and OYI CCPs, as well as the variety 'Capo', achieved slopes below 1, which parallel their low $EV_i$ values, indicating low yield variation over all environments and more specific adaptation to unfavorable environments (Table 1a).

In the AMMI model, based on the 10 experimental years under organic management, the main effect of environment (experimental years) contributed the greatest share to yield variance relative to the total SS of main effects (Model = Env + Gen + Env × Gen) (91 %) (Table 2a), while experimental entries (Gen.) and GEI accounted for 2% and 6%, respectively. The larger GEI effect, in comparison to the genotype effect, indicates the presence of entries particularly suited to specific environments. Removing 'Achat' from the AMMI analysis did not change the results of the ANOVA for grain yield (Table A2). The main effect of environment (Env.) still contributed the greatest share to yield variance relative to the total SS of main effects (92%), whilst experimental entries and GEI accounted for 1.4% and 5.5%, respectively (Table A2).

**Table 2.** Analysis of variance for grain yield of the (a) organic CCP entries and reference varieties (Gen.) across 10 experimental years (Env.) and (b) conventional CCP entries and reference variety across eight experimental years used for the AMMI analysis. The percentual contribution of the main effects relative to the total Sum of Squares (SS) for the model (Model = Env + Gen + Env × Gen) are given.

| (a) Organic CCPs and Reference Varieties | | | | | (b) Conventional CCPs and Reference Variety | | | | |
|---|---|---|---|---|---|---|---|---|---|
| | Df | SS | % of Model SS | % of GEI SS | | Df | SS | % of Model SS | % of GEI SS |
| Model | 87 | 234.7 | | | Model | 61 | 61.8 | | |
| Env. (E) | 9 | 214.4 | 91.3 | | Env. (E) | 7 | 45.0 | 73.0 | |
| Rep(Env) | 8 | 1.8 | 0.8 | | Rep(Env) | 6 | 0.5 | 0.7 | |
| Entry (G) | 7 | 4.0 | 1.7 | | Entry (G) | 6 | 6.1 | 9.9 | |
| GEI | 63 | 14.5 | 6.2 | 6.2 | GEI | 42 | 10.1 | 16.4 | 16.4 |

The AMMI2 Biplot (Figure 1a) accounted for 68% of the yield variation under organic conditions and the CCPs OYQI, OYQII, and OQI were found to be the most stable entries in terms of dynamic stability (no strong interaction with a specific environment), indicated by their low ASV scores (0.24, 0.30, and 0.55, respectively) and proximity to the center of the axes (Table 1a, Figure 1a). Despite a large difference in mean yield between the two experimental years of 2010/2011 (mean yield 7.4 t/ha) and 2013/2014 (mean yield 3.1 t/ha), these are situated near the center of the axes because the varieties differentiated little for yield (Table A3). The reference variety 'Achat' was the least stable in terms of ASV values (1.71, Table 1a), but performed particularly well in 2011/2012 after the bare frost (Figure 1a), in which it also achieved the highest yield (5.2 t/ha). In that year, OYI yielded lowest (3.4 t/ha) (Table A3) due to winter kill [43] and is therefore located in the opposite direction of the 2011/12 vector. 'Achat' is located opposite the vectors of the two very dry years. In terms of $YS_i$, combining both mean yield and ASV ranking, the OYQII CCP achieved the best $YS_i$ value (smallest value), followed by OYQI, OYII and the reference variety 'Achat'. However, both 'Achat' and the OYII CCP achieved higher ASV values, indicating lower general adaptation and suitability to specific environments (static stability) (Table 1a, Figure 1a). Whilst 'Achat' yielded highest in the experimental year 2011/2012 (bare frost), OYII achieved the highest yield in the wettest season 2007/2008 and the two driest seasons 2014/2015 and 2017/2018 (Table A3). Taking all stability indices into account, the OYQII CCP achieved the greatest number of stability values ranked within the top three, followed by the OYQI and OYII CCPs and the reference variety 'Achat' (Table 1a).

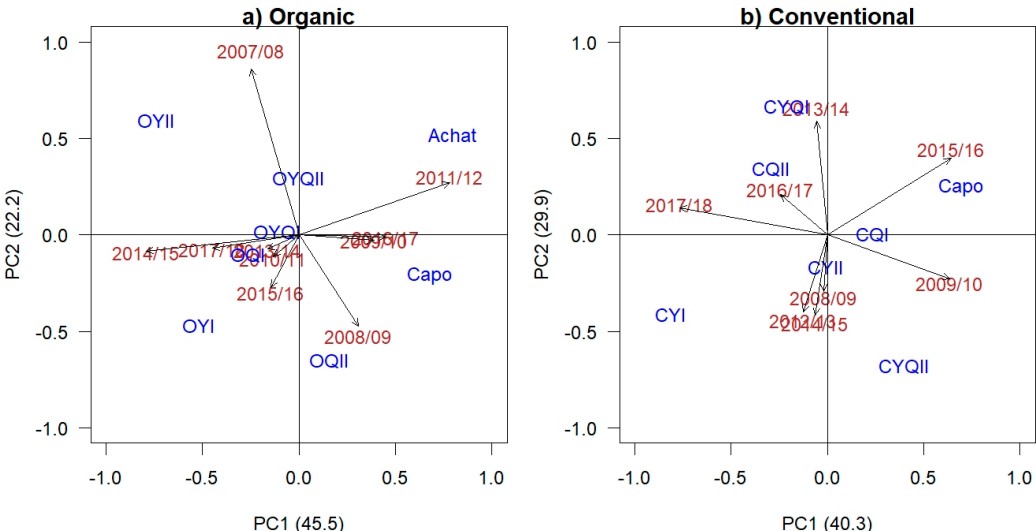

**Figure 1.** AMMI2 Biplot of the first two AMMI interaction (PCA) scores for (**a**) the organic and (**b**) the conventional CCP entries and reference varieties (blue) in 10 and eight experimental years (red), respectively.

### 3.3. Conventional CCPs and Reference Variety

Under conventional management, the reference variety 'Capo' and the CYQI CCP achieved the greatest grain yields over all experimental years with 5.7 and 5.4 t/ha, respectively (Table 1b). The lowest yields were achieved by CQI (4.9 t/ha) and CQII (5.0 t/ha). CQI (0.41) and CYQI (0.43) had the lowest $EV_i$ values indicating the highest stability in the static sense, whilst the CYI CCP had the highest variation in grain yield over the experimental years, indicated by its higher $EV_i$ value of 1.24 (Table 1b). 'Capo' and CYQI obtained the highest yield reliability score ($I_i$) with 5.22 and 4.97, respectively, whilst the CYI CCP the poorest with 4.47 (Table 1b), indicating both a low mean grain yield and high yield variation across multiple environments for this experimental entry. The best values for the dynamic stability measure ($W^2$) (lowest values) were found in the CYII (0.07) and CQI (0.32) CCPs, indicating that these CCPs did not differ greatly in their mean yield performance in relation to the mean environmental yield over the experimental years. This is also reflected in their position near the center in Figure 1b. The CYI CCP and 'Capo' had the highest values for $W^2$ (lower stability in the dynamic sense), tending towards greater GEI as reported by the greater contribution of GEI to $P_i$, as well as higher ASV values (Table 1b). However, the CYI reacted particularly poorly to the conditions in 2009/2010 and 2015/2016 (3.9 and 3.6 t/ha, respectively, Table A3), while 'Capo' yielded particularly well in these years (5.1 and 5.4 t/ha, respectively, Table A3). Therefore, they are located opposite one another in Figure 1b.

The CYII and CYQII CCPs tended towards a slope ($b_i$) of 1, as did the CQII CCP (Table 1b). The reference variety 'Capo' and the CQI and CYQI CCPs had $b_i$ values below 1, indicating adaptation to less favorable environments, particularly CYQI (0.78), which also achieved the highest mean yield of the conventional CCPs (Table 1b). The CYI CCP showed a very strong adaptation to more favorable environments with a slope of 1.45; however, this CCP indicates general instability with poor values for both $EV_i$ and $W^2$, as well as a low mean yield (Table 1b). The reference variety 'Capo' achieved the lowest $P_i$ value indicating its greater yield superiority in comparison to the other entries. However, the GEI contribution to $P_i$ was 84% (Table 1b). The CYII CCP (127) achieved the best $P_i$ values amongst the CCPs. In contrast to 'Capo', the GEI contribution of this CCP was only 24%, indicating a more general adaptation, also confirmed by the lower $W^2$, ASV, and MSE values (Table 1b). The CYQI and CYQII CCPs tended towards similar $P_i$ values (214 and 240, respectively), followed by the CQ CCPs. The CYI CCP had the poorest $P_i$ value (401) with a moderate contribution of the GEI (43%) to the superiority index value.

The AMMI model, based on a dataset of eight experimental years for the conventional entries, indicated that the main effect of environment (experimental years) contributed the greatest share to yield variance relative to the total SS of main effects (Model = Env + Gen + Env × Gen) (73%) (Table 2b). Experimental entries (Genotypes) and GEI accounted for 10% and 16%, respectively, with the greater contribution of GEI to yield variance indicating the presence of entries well suited to specific environments. The AMMI2 Biplot (Figure 1b) accounted for 70.2% of the yield variation, with the CYII, CQI, and CQII CCPs achieving the lowest ASV scores (0.17, 0.31, and 0.53, respectively) and positioned close to the center of the axes (Table 1b, Figure 1b). The two experimental years of 2008/2009 and 2016/2017 indicated the lowest yield variation among entries in comparison to all included years and are therefore also situated in close proximity to the center of the axes. The reference variety 'Capo' and the CYI CCP were the least stable in terms of ASV values (0.97 and 1.71, respectively). The best $YS_i$ value (smallest value) was achieved by the CYII CCP, followed by CYQI and the reference variety 'Capo' (Table 1b), indicating that these three entries ranked well both in terms of yield, as well as in terms of dynamic stability (general adaptation to a range of environments). The CYII CCP achieved the highest number of stability indices ranked in the top three (8), followed by CYQI (5), with the CQI CCP and the reference variety 'Capo' both achieving four stability indices within the top three (Table 1b).

### 3.4. Association between Stability Parameters under Differing Management Systems

A significant positive correlation was found between grain yield ($GY$) and the yield reliability index $I_i$ (index combining both mean yield and $EV_i$), whilst a significant negative correlation was found between grain yield ($GY$) and $EV_i$ (Table 3). $EV_i$ also indicated a significant negative correlation with $I_i$, as well as a significant positive correlation with $b_i$ (slope). $W^2$ (ecovalence) was significantly correlated (positive) with both the stability indices MSE and ASV, both indices representing the dynamic stability concept. ASV and MSE were also strongly significantly correlated (0.84), with the stability index ASV positively correlating with $YS_i$ (a stability index combing both grain yield and ASV) (Table 3).

**Table 3.** Spearman's correlation coefficients among yield and stability indices for all experimental entries under both management systems.

|        | $GY$ | $EV_i$ | $W^2$ | $I_i$ | $b_i$ | MSE | ASV | $YS_i$ | $P_i$ |
|--------|------|--------|-------|-------|-------|-----|-----|--------|-------|
| $GY$   | -    | −0.65 ** | 0.00 | 0.97 *** | −0.18 | 0.09 | −0.01 | −0.43 | −0.44 |
| $EV_i$ |      | -      | 0.43  | −0.65 * | 0.64 ** | 0.30 | 0.35 | 0.11 | −0.17 |
| $W^2$  |      |        | -     | −0.08 | 0.17 | 0.85 *** | 0.95 *** | 0.47 | −0.10 |
| $I_i$  |      |        |       | -     | −0.22 | 0.09 | −0.07 | −0.51 | −0.48 |
| $b_i$  |      |        |       |       | -     | −0.14 | 0.06 | −0.04 | −0.14 |
| MSE    |      |        |       |       |       | -   | 0.84 *** | 0.23 | −0.30 |
| ASV    |      |        |       |       |       |     | -   | 0.55 * | −0.08 |
| $YS_i$ |      |        |       |       |       |     |     | -      | 0.72 ** |

$* \, p < 0.05, ** \, p < 0.01, *** \, p < 0.001$

$YS_i$ was also found to significantly correlate with the superiority index $P_i$ (0.72), a stability index with a strong focus on identifying experimental entries with high yielding capacity. There tended to be moderate correlations between a number of yield stability parameters, although these were not statistically significant. A moderate correlation found between $EV_i$ and $W^2$ (both differing stability concepts), as well as between $EV_i$ and MSE and ASV (0.30 and 0.35, respectively), indicating a certain level of independence, however, some overlap of these stability indices (Table 3).

## 4. Discussion

Under organic conditions, on average, the wheat CCPs yielded comparably to the pure line varieties 'Achat' and 'Capo' over the 10 experimental years. In contrast, under conventional conditions, over the eight experimental years, the mean yield of 'Capo' was statistically significantly higher than that of the CCPs. Yielding ability of the CCPs was affected by their genetic background with the CCQ

populations usually yielding lowest and the CCY and CCYQ yielding higher in both systems. The only exception was the conventional CYI population that yielded significantly less than the CYII population. Lower yields of the CCQ populations were expected based on their genetic background for high baking quality, rather than yielding ability. Similar results for lower yields of the CCQ populations in comparison to the other CCP groups have been reported by Döring et al. [40] and Brumlop et al. [43].

Under organic management, no particular differences for yield stability between the parallel CCPs were found. The OYQII population achieved above average yields and had the greatest stability (highest number of top ranks for the various stability parameters) with low GEI (greater dynamic stability), indicated by low $W^2$, MSE, and ASV scores, as well as a higher percentual contribution of genetic deviation to $P_i$. Whilst the regression coefficient ($b_i$) was included as a stability parameter in this study, Becker and Leon [20] report that due to a number of criticisms of the measure, it is best used as an additional evaluation tool with which to describe a genotype's adaptation and response to unfavorable environments. Bearing this in mind, the OYQII CCP had a slope of 1.07, like 'Achat', but the lowest MSE value and as such indicated good stability and adaptation to all environments [27]. The CCYQ populations are the most genetically diverse due to their parental makeup (20 parental varieties) and as such have the advantage of a larger parent set with both high yielding and high baking quality characteristics. Döring et al. [40] found similar results when comparing the grain and protein stability of the same CCPs under differing management systems. Both the CCY and CCYQ populations tended towards lower (better) $EV_i$ values for grain yield in comparison to the mean and mixture of their respective parental varieties across both organic and conventional environments [40]. Greater yield stability of barley CCPs over a number of reference varieties has also been reported by Soliman and Allard [36], indicating the potential use of CCPs to create cultivars characterized by low GEI. Creissen et al. [38] working on winter barley mixtures, found increased yield stability of two barley mixtures over two pure lines varieties for the dynamic stability measure $W^2$. The reference variety 'Capo' in our study also displayed a high degree of yield stability, tending towards adaptation to less favorable environments (greater GEI) and high stability in terms of environmental variance ($EV_i$, static stability) and $I_i$, particularly in comparison to the CCPs under conventional management.

Under conventional management, the CYII CCP was found to be the most stable genotype, followed by the CYQI and CQI CCPs and the reference variety 'Capo'. The CCY and CCYQ CCPs were the same CCPs reported by Döring et al. [40] for their better yield stability in comparison to their respective parental variety mixtures and means. A comprehensive analysis of the parental varieties by Jones et al. [46] reports on the advantage of the Y parental set under conventional management due to their later release date (more modern varieties) and better adaptation to agricultural systems with higher N inputs. Thus, it appears that regular fertilization application with mineral N to the top soil selected for higher root density and thinner roots in that zone, a property most pronounced in the Y genetics. Under organic management, selection was for longer seminal roots and more even root distribution across depths [68]. This could explain why under conventional management, the population with the high yielding genetic background, CYII CCP, was the most stable while under organic management the broader genetic background of the OYQ CCPs was most stable. The modern genetic base found in the CYII population and the CYQI (crossed with yield parental set) may have given them a yield advantage, as well as a yield stability advantage under conventional conditions. The high yielding parental varieties were found to yield better under conventional management and were less suitable for organic systems characterized by higher abiotic and biotic stress [46].

As pointed out above, the CYI CCP yielded significantly lower than the CYII population (Table 1b). Additionally, the CYI CCP showed greater instability in both the static and dynamic sense, suggesting potentially divergent evolutionary trajectories that were also confirmed by the studies on seedling traits [68]. The slope of the regression analysis for CYI was comparably steep (1.45), indicating below average stability and an adaptation to more favorable conditions. Similar differential trajectories for mean yield were also found between the two parallel CY populations, with the CYI achieving both one of the highest and the lowest yields across the eight experimental seasons analyzed (Figure 1b,

Table A3). Goldringer et al. [69,70] reported on the divergent evolution of French wheat CCPs subjected to differing environments, whilst the management history of the conventionally grown CCPs and reference variety also included a location change to another field in 2013/2014, which may have contributed to the differential trajectories of the two conventionally managed parallel CCY populations.

The mechanisms of compensation, complementation, and facilitation have been given as ecological processes, which may increase stability in genetically diverse populations [2,38,40,71]. Although the exact mechanisms that inferred better stability to the OYQII CCP in particular are unknown, the German CCPs showed remarkable resilience and compensation effects in the experimental year of 2011/2012 [43], when the severe black frost in February 2012, followed by a drought described above occurred. There was a high degree of winterkill in in the CCP parental varieties, which had been grown in plots neighboring the populations, indicating a generally poor adaptation to such conditions compared to the reference varieties 'Achat' and 'Capo'. Although only four of the parental varieties (mainly from the Q CCPs) survived reasonably well due to good winter hardiness, the CCPs, characterized by greater genetic diversity, recovered sufficiently enough to achieve a mean trial yield of 4.2 t/ha. No yields could be determined for the parental varieties as the trial had to be terminated due to the winter kill [43]. Creissen et al. [72] showed that in genotype mixtures in Arabidopsis, seed yield stability in the mixtures was maintained through the mechanism of compensation. This compensatory effect for seed yield was pronounced under abiotic stress, and could have provided the OYQ populations, which are more genetically diverse, an advantage under the more challenging conditions of the organic management system.

According to Annicchiarico and Filippi [14] and Lammerts van Bueren and Myers [3], organic systems are considered more challenging due to higher abiotic and biotic pressures. In organic systems, there are few immediate control options when dealing with insect pests and pathogens. Additionally, abiotic stresses such as low N availability under organic conditions, particularly in early spring, make the growing conditions in organic and low-input systems more challenging [73,74]. While no pathogen or insect control was applied in both management systems described here, nutrient dynamics were likely different in the two systems. Climatic conditions in Neu-Eichenberg over the last few years have been characterized by more frequent drought periods between late winter and early spring. This has resulted in lower soil biological activity and thus N mineralization during this critical growth period and likely led to overall more stressful abiotic conditions for wheat under organic management. This is reflected by the considerably higher percentage of the variance due to environment under organic conditions (91%) compared to conventional (73%) and in turn higher GEI for the conventional system (Table 2).

Slower mineralization and lower N inputs in the organic system, particularly under drought stress, leads to earlier senescence. In addition to early maturation, which leaves plants unable to make use of later rain periods for grain filling, N sources during grain filling under organic management are often limited [75,76]. These greater challenges found in the organic system may be mitigated by greater intra-specific diversity, as is the case with the OYQ CCPs and greater yield stability. This was clearly visible in this study where, depending on the growing system in certain years, the entries differed strongly in their yield performance, while in others they all behaved similarly. For example, the year 2013/2014 was extremely difficult under organic conditions resulting in poor yields between 2.6 and 3.4 t/ha with no significant differences among entries. This was due to a very warm and dry late winter resulting in poor and late mineralization of nutrients. Similarly, in 2010/2011, entries did not yield differentially, but the weather was favorable with soils coming through the winter with adequate water content. The warm April and May allowed for timely nutrient mineralization and yields ranged from 7.2 and 7.8 t/ha. In contrast, under conventional conditions when provided with mineral nitrogen fertilizer, nutrients were less limiting and yields in 2013/2014 ranged from 4.2 to 5.5 t/ha. Unfortunately, for 2010/2011, no yield data are available for the conventional site. Growing system also affected the pure line 'Capo'. For example, under organic management, 'Capo' did particularly well in 2009/2010

and in 2016/2017 and poorest in 2015/2016. Under conventional management, it was the best performing entry in 2015/2016.

The significantly lower yields of the conventionally managed CCPs in comparison to the pure line variety 'Capo' highlights the importance of introducing new genetic material into the CCPs in order to improve yielding ability of heterogeneous populations, but also to improve competitiveness under favorable conventional conditions. The parents of the CCPs were chosen specifically for their performance under low-input conditions in Europe [40,43,46] and as such are most likely unable to take advantage of the higher N-inputs found in current conventional systems. Therefore, the addition of elite pure lines into CCPs, as proposed by Döring et al. [40], may help to improve yields, particularly in conventional systems. Additionally, Knapp et al. [77] reported on the "reversion to wild type" and loss of alleles related to dwarfing in wheat CCPs from the UK, but based on the same genetic background as the CCPs in this study. This is particularly relevant as the trade-off between competitive ability in the form of plant height has been negatively correlated with agronomic performance [40,69,77]. This competitive ability may result in lower yields due to greater investment in biomass rather than in kernel number and weight, and can increase the risk of lodging, further reducing grain yields [40,78]. For these reasons, the application of plant growth regulators to reduce height is recommended for CCPs under conventional management.

The significant positive correlation between grain yield and the yield reliability index $I_i$ indicates no trade-off between yield and stability, as greater yield reliability for $I_i$ is indicated by increasing values. The negative correlation between grain yield and superiority index $P_i$ is in line with this, as lower values indicate greater stability and yielding superiority for $P_i$. Therefore, both $P_i$ and $I_i$ could be used as stability measures that select for both stable and high yielding genotypes. A significant negative correlation between grain yield and $EV_i$ indicates that a potential tradeoff between stability (in the static sense) and grain yield was not present. However, low correlations between grain yield and the dynamic stability indices $W^2$, MSE and ASV, indicate the independence of these stability parameters from grain yield and the potential trade-off between better grain yield and stability, as well as providing information on yield stability that cannot be deduced from yield data alone [79]. This, however, illustrates the advantages of using a number of stability parameters in order to ensure adequate criteria for genotypes in specific environments [80]. There were moderate to high correlations between a number of parameters based on the same stability concept, such as between $b_i$, which can indicate both static and dynamic stability, and $EV_i$, as well as between the dynamic stability indices $W^2$, ASV, and MSE. Similar results have been reported by a number of researchers [79–82]. Importantly, not all stability parameters were correlated, indicating a differential stability evaluation and independence, which may aid selection depending on the genotype characteristics desired.

## 5. Conclusions

The wider genetic base of the CCPs in general provided for good yield stability. Under organic management, the YQ CCPs with their wider genetic base tended towards better overall yield stability, particularly in the dynamic sense, and above average yields in comparison to the other CCP groups and reference varieties. This wider genetic background tended to be advantageous in the buffering of higher abiotic and biotic stresses, which characterize organic and low-input systems. In terms of static stability ($EV_i$), the OQI and OYI CCPs showed the greatest stability, indicating the importance of CCP choice depending on the environment in order to take advantage of specific GEIs. Thus, the importance of parental selection in the adaptation and suitability of CCPs to specific environments cannot be understated. The challenging and often marginal environments of organic and low-input agriculture necessitate the inclusion of stability parameters that quantify static stability and the importance of multiple long-term trial environments including a number of reference varieties is evident. Although overall yield performance of the conventionally managed CCPs was lower than that of the reference variety 'Capo', overall yield stability was better than that of 'Capo', particularly of the high yielding CYII CCP. This indicates the advantage of the more modern parental genetics present in the Y CCPs

able to make better use of the higher N-input and the advantage of integrated genetic diversity. As the breeding of heterogeneous crop populations is an approach supported mainly by organic farmers, researchers and breeders, little effort or demand has pushed for the creation of genetically diverse populations better suited to conventional systems. Improving CCPs for conventional systems may also involve the admixing of elite breeding lines as suggested by Döring et al. [40].

Although scientific research and legislative ruling have been supportive, a major hindrance to the establishment of CCPs in the market is the lack of experience with diversified materials along the processing chain. Future research must focus on knowledge generation on the potential opportunities and barriers value chain actors may face through their adoption of CCPs. This knowledge will enable multi-actor-driven actions, as well as policy amendments or additions in order to provide possible solutions to support and amend the strategic framework ensuring wider CCP acceptance. Overall, CCPs based on materials adapted to a given management system (organic or conventional, but also others such as irrigated or dry land, for example) provide an interesting and promising alternative to pure line varieties that may help to address climate change, decentralize the breeding process, increase stability and diversity within the agricultural landscape, and provide novel and diverse food products.

**Author Contributions:** Conceptualization of the study, O.D.W. and M.R.F.; methodology, O.D.W. and M.R.F.; formal analysis, O.D.W.; investigation, O.D.W.; data curation, O.D.W. and M.R.F.; writing—original draft preparation, O.D.W. and M.R.F.; writing—review and editing, O.D.W. and M.R.F.; supervision, M.R.F.; project administration, O.D.W. and M.R.F.; funding acquisition, M.R.F.

**Funding:** This research was partly funded through the "Zentrale Forschungsförderung" University of Kassel, "Bundesprogramm Ökologischer Landbau und andere Formen nachhaltiger Landwirtschaft", Project Nr. 2812OE021 in the framework of CORE Organic II and through the INSUSFAR (INovative approaches to optimize genetic diversity for SUStainable FARming systems of the future) Project, Project Nr. FKZ 031A350C, financed by the "Bundesministerium für Bildung und Forschung" in the framework of the IPAS (Innovative Pflanzenzüchtung im Anbausystem) Initiative.

**Acknowledgments:** We thank Sarah Brumlop and Andreas Butz, and numerous BSc and MSc students for their help in the work on the heterogeneous wheat populations in Neu-Eichenberg and Günter Kellner, Rainer Wedemeyer, Joachim Deckers and Sven Heinrich for technical support.

**Conflicts of Interest:** The authors declare no conflict of interest. The funders had no role in the design of the study; in the collection, analyses, or interpretation of data; in the writing of the manuscript, or in the decision to publish the results.

# Appendix A

**Table A1.** Mean monthly and overall mean temperature (°C) and total monthly and overall total precipitation (mm) per experimental season from 2005/2006 to 2017/2018. The long-term means (1971–2000) for temperature and precipitation are also given.

| Experimental Season | Sept °C | Sept mm | Oct °C | Oct mm | Nov °C | Nov mm | Dec °C | Dec mm | Jan °C | Jan mm | Feb °C | Feb mm | Mar °C | Mar mm | April °C | April mm | May °C | May mm | June °C | June mm | July °C | July mm | Aug °C | Aug mm | Mean °C | Mean mm |
|---|---|---|---|---|---|---|---|---|---|---|---|---|---|---|---|---|---|---|---|---|---|---|---|---|---|---|
| 2005/2006 | 14.9 | 28 | 11.3 | 20 | 5.2 | 38 | 1.8 | 56 | −2.4 | 21 | −0.5 | 40 | 1.5 | 56 | 8.1 | 38 | 12.7 | 84 | 16.3 | 28 | 21.2 | 59 | 15.6 | 73 | 8.8 | 541 |
| 2006/2007 | 16.8 | 18 | 12.7 | 49 | 7.8 | 41 | 5.2 | 48 | 5.3 | 99 | 4.4 | 57 | 6.5 | 59 | 10.7 | 5 | 13.8 | 103 | 17.2 | 117 | 17.2 | 77 | 16.4 | 78 | 11.2 | 751 |
| 2007/2008 | 12.8 | 128 | 8.5 | 23 | 4.0 | 106 | 1.7 | 69 | 4.5 | 70 | 4.0 | 25 | 4.4 | 82 | 7.7 | 73 | 14.5 | 16 | 17.1 | 91 | 18.3 | 56 | 17.9 | 46 | 9.6 | 785 |
| 2008/2009 | 12.7 | 48 | 9.2 | 66 | 5.2 | 54 | 0.3 | 61 | −2.4 | 11 | 1.1 | 33 | 4.5 | 75 | 12.3 | 33 | 13.9 | 78 | 14.7 | 51 | 18.6 | 81 | 18.6 | 69 | 9.1 | 660 |
| 2009/2010 | 14.6 | 77 | 8.3 | 67 | 8.0 | 81 | 0.3 | 96 | −3.8 | 10 | −0.3 | 42 | 4.6 | 71 | 9.2 | 19 | 10.6 | 89 | 16.4 | 46 | 20.7 | 48 | 16.7 | 147 | 8.8 | 793 |
| 2010/2011 | 12.5 | 86 | 8.3 | 29 | 4.8 | 87 | −4.1 | 43 | 1.3 | 49 | 0.4 | 29 | 3.5 | 9 | 12.7 | 37 | 15.6 | 18 | 17.3 | 90 | 16.5 | 43 | 18.4 | 105 | 8.9 | 625 |
| 2011/2012 | 15.4 | 41 | 9.8 | 42 | 4.8 | 2 | 4.1 | 111 | 2.4 | 121 | −2.3 | 24 | 7.5 | 15 | 8.4 | 35 | 14.6 | 61 | 14.9 | 127 | 17.3 | 140 | 18.9 | 59 | 9.7 | 778 |
| 2012/2013 | 13.8 | 44 | 8.8 | 42 | 5.2 | 33 | 2.1 | 90 | −0.1 | 53 | −0.6 | 51 | −0.6 | 25 | 8.4 | 33 | 11.6 | 146 | 15.2 | 26 | 19.0 | 36 | 18.4 | 37 | 8.4 | 616 |
| 2013/2014 | 13.8 | 62 | 11.2 | 81 | 4.8 | 72 | 4.8 | 37 | 2.8 | 38 | 5.3 | 16 | 7.3 | 11 | 11.5 | 29 | 12.4 | 96 | 15.2 | 37 | 19.0 | 125 | 16.0 | 79 | 10.3 | 683 |
| 2014/2015 | 15.0 | 37 | 12.3 | 45 | 7.0 | 15 | 2.7 | 36 | 2.7 | 44 | 1.3 | 18 | 5.1 | 50 | 8.6 | 36 | 12.3 | 25 | 15.5 | 28 | 19.0 | 95 | 19.6 | 160 | 10.1 | 589 |
| 2015/2016 | 12.8 | 51 | 8.5 | 43 | 8.8 | 89 | 7.6 | 29 | 1.7 | 40 | 3.5 | 83 | 4.2 | 45 | 8.0 | 47 | 13.8 | 42 | 17.0 | 99 | 18.6 | 65 | 18.0 | 24 | 10.2 | 657 |
| 2016/2017 | 17.5 | 21 | 8.9 | 71 | 4.2 | 33 | 2.4 | 16 | −2.1 | 36 | 3.4 | 37 | 7.8 | 39 | 7.5 | 37 | 14.4 | 32 | 17.5 | 60 | 18.1 | 183 | 17.7 | 124 | 9.8 | 689 |
| 2017/2018 | 13.0 | 35 | 11.9 | 61 | 5.8 | 73 | 3.6 | 47 | 4.2 | 80 | −1.8 | 9 | 2.7 | 43 | 12.8 | 29 | 15.6 | 33 | 17.6 | 15 | 20.6 | 15 | 20.2 | 31 | 10.5 | 471 |
| **Mean** | 14.3 | 52 | 10.0 | 49 | 5.8 | 56 | 2.5 | 57 | 1.0 | 52 | 1.4 | 36 | 4.5 | 45 | 9.7 | 35 | 13.5 | 63 | 16.3 | 63 | 18.8 | 79 | 17.9 | 79 | 9.6 | 664 |
| **Long-term mean (1971–2000)** | 13.6 | 52 | 9.2 | 44 | 4.7 | 51 | 2.3 | 58 | 1.0 | 49 | 1.4 | 36 | 4.9 | 49 | 8.1 | 43 | 12.9 | 58 | 15.5 | 74 | 17.4 | 59 | 17.3 | 55 | 9.0 | 628 |

**Table A2.** Analysis of variance for grain yield of the organic CCP entries and reference variety 'Capo' (Gen.) across 10 experimental years (Env.) used for the AMMI analysis. The percentual contribution of the main effects relative to the total Sum of Squares (SS) for the model (Model = Env + Gen + Env × Gen) are given.

| | Df | SS | % of Model SS | % of GEI SS |
|---|---|---|---|---|
| Model | 77 | 201.4 | | |
| Env. (E) | 9 | 185.5 | 92.1 | |
| Rep(Env) | 8 | 1.9 | 0.9 | |
| Entry (G) | 6 | 2.9 | 1.4 | |
| GEI | 54 | 11.1 | 5.5 | 5.5 |

**Table A3.** Estimated mean yields of the (a) organic CCPs and reference varieties over 10 experimental seasons and (b) conventional CCPs. and reference varieties over eight experimental years. Differing letters indicate significant differences at $p < 0.05$ within each experimental year and management system (Linear mixed effects model with pairwise comparison and Holm correction).

**(a) Organic CCPs and reference varieties**

| Entries | 2007/2008 | | 2008/2009 | 2009/2010 | | 2010/2011 | 2011/2012 | | 2013/2014 | 2014/2015 | 2015/2016 | | 2016/2017 | 2017/2018 | |
|---|---|---|---|---|---|---|---|---|---|---|---|---|---|---|---|
| Achat | 6.7 | ab | 5.3 | 6.5 | a | 7.4 | 5.2 | a | 2.9 | 4.2 | 4.3 | ab | 6.4 | 4.3 | cd |
| Capo | 5.7 | ab | 5.5 | 6.6 | a | 7.3 | 4.8 | ab | 3.5 | 4.2 | 3.9 | c | 5.9 | 4.3 | cd |
| OQI | 5.6 | ab | 4.8 | 5.1 | b | 7.2 | 4.2 | bc | 2.6 | 4.8 | 3.9 | c | 5.4 | 4.4 | c |
| OQII | 5.3 | b | 5.2 | 6.1 | ab | 7.4 | 3.9 | cd | 2.7 | 4.6 | 4.4 | a | 5.7 | 4.0 | e |
| OYI | 5.8 | ab | 5.2 | 5.9 | ab | 7.2 | 3.4 | d | 3.2 | 5.0 | 4.3 | abc | 5.5 | 4.7 | b |
| OYII | 7.0 | a | 4.7 | 6.1 | ab | 7.5 | 4.0 | cd | 3.4 | 5.4 | 4.2 | abc | 5.4 | 4.9 | a |
| OYQI | 6.3 | ab | 4.9 | 6.1 | ab | 7.8 | 3.9 | cd | 3.4 | 4.3 | 4.6 | a | 5.6 | 4.3 | c |
| OYQII | 6.5 | ab | 5.0 | 6.4 | a | 7.5 | 4.3 | bc | 3.3 | 5.1 | 4.0 | bc | 6.1 | 4.2 | d |

**(b) Conventional CCPs and Reference Variety**

| Entries | 2008/2009 | | 2009/2010 | | 2012/2013 | 2013/2014 | | 2014/2015 | | 2015/2016 | | 2016/2017 | | 2017/2018 | |
|---|---|---|---|---|---|---|---|---|---|---|---|---|---|---|---|
| Capo | 5.9 | a | 5.1 | a | 6.2 | 5.5 | a | 6.6 | a | 5.4 | a | 6.9 | a | 4.4 | d |
| CQI | 5.0 | cd | 4.3 | bc | 5.4 | 4.6 | cd | 6.0 | ab | 4.4 | b | 5.6 | c | 4.5 | cd |
| CQII | 4.9 | d | 3.9 | c | 5.5 | 4.5 | cd | 6.0 | ab | 4.6 | b | 6.3 | b | 5.1 | ab |
| CYI | 5.6 | abc | 3.9 | c | 6.1 | 4.7 | cd | 6.4 | ab | 3.6 | c | 6.4 | ab | 5.1 | abc |
| CYII | 5.6 | ab | 4.8 | ab | 6.0 | 4.9 | bc | 6.3 | ab | 4.4 | b | 6.4 | ab | 5.0 | abc |
| CYQI | 5.3 | bcd | 4.7 | ab | 5.7 | 5.4 | ab | 5.7 | b | 4.5 | b | 6.7 | ab | 5.4 | a |
| CYQII | 5.6 | abc | 5.2 | a | 6.2 | 4.2 | d | 6.3 | ab | 4.5 | b | 6.2 | b | 4.7 | bcd |

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
