# Peer review of "Heterogeneous Winter Wheat Populations Differ in Yield Stability Depending on their Genetic Background and Management System"

_sustainability, doi:10.3390/su11216172_

Round 1

Reviewer 1 Report

The manuscript fits within the scope of the journal. The manuscript is interesting and the idea is very nice.  The title is clear and it is adequate to the content of the article. The study methods are explained clearly. The author’s work on discussing achieved results is appreciated.

I have some recommendations for authors:

- Pay more attention to text editing rules. The lack of space between words, or between propositions is a problem. Please check all the text. For example see line 32, 51….

- Please include some future research directions.

- Please restructure the conclusions. Just refer to your results. In the conclusions chapter bibliographic sources should not be found.

Author Response

Thank you very much for your review of our manuscript. Please see the attachment for our reply to your comments.

Reviewer 2 Report

The manuscript describes the yield performance and stability of wheat composite cross populations (CCPs) under both organic and conventional management. Although I understand the importance of a much amount of data for 10 years and statistical analysis, there is doubt about the validity of the conclusion.

In "Conclusions", the manuscript describes that "Under organic management, the YQ CCPs with their wider genetic base tended towards better yield stability" (Lines 565 and 566). Indeed, the lowest ecovalence values, which indicate dynamic stability, is exhibited in YQI under organic management.  However, the parameter of static stability, environmental variance, shows that other CCP has yield stability. Does this result support the conclusion?"

The last sentence of "Conclusion" is inconsistent with the results of this manuscript. This manuscript shows that "there are no significant yield differences between organically managed CCPs and reference varieties" (Lines 278 and 279). This does not support the description of "CCPs show great potential not only in terms of yield performance" (Line 582).

From these reasons, I cannot recommend the acceptance of this manuscript.

Author Response

Thank you very much for the review of our manuscript. Please see the attachment for our reply to your comments.

Reviewer 3 Report

Review of “Heterogeneous Winter Wheat Populations Differ in Yield Stability Depending on their Genetic Background and Management System” by Weedon and Finckh for MDPI Sustainability (614896).

This paper reports the findings from the analysis of the yield performance of heterogeneous winter wheat populations of variable genetic backgrounds over up to 10 years under two crop management systems. The study was well executed, the stability analysis was appropriated, and the findings are useful for understanding the relationships between genetic diversity and yield stability under changing environments. I would recommend for publication, with some consideration of my minor corrections below to improve

Comments

Line77: The High genetic Diversity? I would suggest to label the variety as ‘Achat’, rather than Achat. Line 185: P=0.75 vs Line 608 p<0.05 Line 422: no difference Line 556: This, however, illustrates

Author Response

Thank you vry much for your review of our manuscript. Please see the attachment for our reply to your comments.

Round 2

Reviewer 2 Report

The revised manuscript has addressed the issues raised in previous review. The manuscript is now suitable for publication.